# Effects of Perceptual Learning on Deprivation Amblyopia in Children with Limbal Dermoid: A Randomized Controlled Trial

**DOI:** 10.3390/jcm11071879

**Published:** 2022-03-28

**Authors:** Jing Zhong, Wei Wang, Jijing Li, Yiyao Wang, Xiaoqing Hu, Lei Feng, Qingqing Ye, Yiming Luo, Zhengyuan Zhu, Jinrong Li, Jin Yuan

**Affiliations:** 1State Key Laboratory of Ophthalmology, Zhongshan Ophthalmic Center, Guangdong Provincial Key Laboratory of Ophthalmology and Visual Science, Sun Yat-Sen University, Guangzhou 510060, China; zhongjing_90@163.com (J.Z.); wangw289@mail.sysu.edu.cn (W.W.); lijj57@mail2.sysu.edu.cn (J.L.); yaoyaoyatou42@163.com (Y.W.); Xiaoqinghu2010@163.com (X.H.); i_ivanfeng@yeah.net (L.F.); yqq0596@163.com (Q.Y.); 2Guangdong Provincial Clinical Research Center for Ocular Diseases, Guangzhou 510060, China; 3Guangzhou LWT Technologies Co., Ltd., Guangzhou 510060, China; luoyiming@eyechina.com.cn; 4Shenzhen CESI Information Technology Co., Ltd., Shenzhen 518100, China; zhuzy@nels.org.cn

**Keywords:** limbal dermoid, contrast sensitivity function, visual acuity, perceptual learning

## Abstract

Limbal dermoid (LD) is a congenital ocular tumor that causes amblyopia and damages visual acuity (VA) and visual function. This study evaluated the therapeutic efficacy of perceptual learning (PL) toward improving contrast sensitivity function (CSF) and VA. A total of 25 children with LD and 25 normal children were compared in terms of CSF and VA. The LD group was further randomly allocated into two arms: nine underwent PL combined with patching and eight underwent patching only; eight patients quit the amblyopia treatment. The primary outcome was the area under log CSF (AULCSF), and the secondary outcome was the best corrected VA (BCVA). The CSF was obviously reduced in the LD group compared with that in the normal group. Moreover, the difference in the changes in the AULCSF between the PL and patching groups after 6 months of training was 0.59 (95% CI: 0.32, 0.86, *p* < 0.001), and the between-group difference in VA at 6 months was −0.30 (95% CI: −0.46, −0.14, *p* < 0.001). Children suffering from LD with amblyopia exhibited CSF deficits and VA loss simultaneously. PL could improve CSF and VA in the amblyopic eye better than patching.

## 1. Introduction

Limbal dermoid (LD) is a congenital benign ocular tumor that affects vision and causes visual abnormalities due to induced corneal astigmatism [1]. Based on a grading system for LDs used for clinical diagnosis, a low-grade dermoid is associated with better vision postoperatively [2]. Although keratoplasty, such as lamellar keratoplasty (LKP) or penetrating keratoplasty (PKP), has been shown to facilitate ocular reconstruction and to significantly improve visual acuity (VA) postoperatively, visual function is largely ignored by parents after visual appearance recovery [3,4]. It has been documented that keratoplasty do not significantly improve objective and subjective visual functions postoperatively, including contrast sensitivity (CS), refractive error, graft clarity, anterior and posterior corneal higher order aberration, and vision-related quality of life [5]. In addition, the evaluation of visual function for LD is performed less frequently, although a subjective visual function assessment has been deemed critical [6].

A large proportion of patients with epibulbar dermoids have been shown to suffer from amblyopia [2], a common postoperative complication in LD patients. The loss of vision is thought to be secondary to abnormal relationships within the neuronal network in the primary visual cortex [7]. Amblyopia is characterized by several functional abnormalities in spatial vision, including reductions in VA, contrast sensitivity function (CSF), and spatial distortion. Traditional treatment for amblyopia is based on penalization of the good eye while optimizing the visual function of the amblyopic eye in childhood [8,9]; however, this treatment is accompanied by problems that are social or emotional in nature, by skin irritation, and by other problems that might affect compliance [10].

Perceptual learning (PL) is a new treatment option involving visual tasks that help improve visual performance, including VA and CS [11,12]. The effects of PL on visual functions in amblyopia have been documented, and PL has been applied in adults with hypermetropic anisometropic amblyopia [11] and in children with amblyopia [10] and presbyopia [13]. Despite its great potential as a novel treatment for amblyopia secondary to epibulbar dermoids both preoperatively and postoperatively, to our best knowledge, no randomized controlled trial (RCT) on PL has ever been conducted.

To augment the current knowledge gap, this study focused on children with LD who underwent LKP and were diagnosed as having amblyopia. It evaluated the VA and CSF of the patients and then compared the therapeutic efficacy of PL and part-time patching toward improving the VA and CSF of the patients.

## 2. Methods

### 2.1. Clinical Demographics

This study included 25 LD children (LD group) and 25 healthy children (N group), and their demographic characteristics are shown in Table 1. The LD group included 12 females (48%) and 13 males (52%) with an average age of 10.20 (8.30, 11.85) years, whereas the N group consisted of 12 females (48%) and 13 males (52%) with an average age of 10.20 ± 1.95 (7.00–13.00) years. The mean sphere measurements for the LD and N groups were −3.86 (−7.75, −0.25) and −4.26 (−5.87, −2.25), respectively, and the cylinder measurements were 4.86 (4.75, 7.75) and 0.30 (0.13, 0.50), respectively; cylinder diopters were evidently higher by 4D in the LD group than in the N group. In the LD group, the average age of the 9 children who participated in the PL training was 9.31 (7.45, 11.65) years and that of the 8 children in the part-time patching group was 10.40 ± 2.00 (7.00–12.00) years; the mean sphere measurements were −2.86 (−8.00, −0.25) and −3.97 (−7.69, −0.31), and the mean cylinder measurements were 4.36 (2.63, 6.63) D and 4.34 ± 2.96 D (1.00–5.50) in the LD and part-time patching group, respectively (Table 1). Eight patients quit the amblyopia treatment due to inability to complete the PL training and due to other reasons.

### 2.2. Study Design

This study was performed in the Corneal Disease Department of Zhongshan Ophthalmic Center, Guangzhou, China, between March 2018 and December 2020. The research was performed according to the tenets of the Declaration of Helsinki. The trial protocol and consent form were reviewed and approved by the Zhongshan Ophthalmic Ethical Committee. The trial was registered at clinicaltrials.gov (PRS, ID NCT03447041). Clinical measurements were performed after a written informed consent was obtained from all participants.

### 2.3. Participants

This study was divided into two stages. In stage 1 (single-center, prospective cross-sectional study), all patients underwent a visual function evaluation test. In stage 2 (RCT), the LD patients were randomly assigned to receive PL training combined with part-time patching or part-time patching therapy alone. The LD eye was selected in the LD group, and the dominant eye was chosen in the N group for further analysis.

In the cross-sectional study, LD children who underwent LKP (LD group) and normal subjects (N group) were continuously recruited from the Corneal Disease Department of Zhongshan Ophthalmic Center (Table 1). The inclusion criteria for the LD group were as follows: (1) diagnosis of LD, underwent LKP, and underwent corneal stitch removal at 1 year postoperatively; (2) aged > 7 years; and (3) diagnosed with amblyopia with a logMAR BCVA of 1.00 or better. The inclusion criteria for the control group were as follows: (1) aged > 7 years, (2) a logMAR BCVA of 0.00 or better, (3) no history of any ocular pathology, and (4) with normal physical and mental health. Patients who are unable to under objective VA measurements or to complete the PL training (e.g., too young or developmentally delayed) were excluded. Those who were lost to follow-up or who came for second opinion only (i.e., only one follow-up visit) and those who had surgical treatment during the study period were excluded from the analysis. Additionally, patients suffering from any other eye diseases (e.g., optic nerve disease, retinal disease, eyelid, or corneal pathology causing deprivation amblyopia) with neurological associations were excluded.

The patients in the LD group underwent monocular and binocular VA assessments and a quick CSF (qCSF) assessment at full optical correction after a routine ophthalmic examination. In the N group, the children were subjected to VA and CSF assessments under two conditions, that is, full refractive correction and full refractive correction plus optical defocus (+1.00 D to +6.00 D positive spherical lens) on the right eye, with the vision of the left eye unaltered [14,15,16,17]. Specifically, optical defocusing was used to simulate monocular blurred BCVA in the LD group. For example, five patients had a logMAR VA of 0.40; accordingly, the vision of five normal subjects was blurred to a logMAR VA of 0.40 using optical defocus to match the vision of the LD patients during the qCSF assessment.

All subjects were screened using a detailed baseline eye examination, wherein the parameters assessed included manifest refraction, BCVA (with the use of the Early Treatment Diabetic Retinopathy Study (ETDRS) logMAR chart), CSF at 1.5, 3, 6, 12, and 18 cycles for each spatial frequency (SF) and intraocular pressure; slit-lamp biomicroscopy and fundus examinations were also conducted. The same protocol was adopted for examinations at baseline and follow-up, and the same measurements were performed by the same examiner using the same device.

The qCSF method included 10 alternative forced-choice digit identification tasks to assess the CSF. Stimuli were displayed on a gamma-corrected 46-inch LCD monitor (Model: NEC LCD P463, Samsung, Suwon City, Korea) with 1920 × 1080 pixel resolution, 50 cd/m^2^ mean luminance, and 60 Hz vertical refresh rate. Thirty trials were conducted, with a total testing time of approximately five minutes. The qCSF data were scored to generate (1) the area under log CSF (AULCSF), a summary descriptor of contrast thresholds in spatial vision, at spatial frequencies of 1, 1.5, 3, 6, 12, and 18 cpd; the higher the AULCSF, the better the CSF function will be; and (2) the cutoff SF, which is the SF acuity corresponding to a perceptual CS at 50%; the higher the cutoff SF, the better the CSF function will be. The AULCSF and cutoff SF values were used for further analysis [16].

### 2.4. Randomization and Other Procedures

In stage 2 (RCT), the children with LD after LKP with amblyopia were randomly allocated into two arms, namely, the PL group (n = 9) and the part-time patching group (n = 8). Single randomization sequences were generated using SPSS version 20 (IBM, Inc., Armonk, NY, USA)) by an independent statistician, who placed the allocation details in sequentially numbered, opaque, sealed envelopes, which were kept from the investigators until the enrolled participants have completed all the baseline assessments. The participants were randomly assigned at a 1:1 ratio to either the PL group or to the patching alone group. Blinding was not applicable given the intervention method employed; however, the technicians and research assistants who were responsible for screening, enrollment, and follow-up measurements were blind to the treatment allocation.

After the randomization (±1 week), follow-up visits were scheduled at 1 week and at 1, 2, 3, 4, 5, and 6 months. The children in the LD group (n = 9) underwent PL for successive 30-min sessions daily (Figure 1B) and were prescribed 2 h of daily patching with an adhesive-style patch (i.e., Coverlet, 3M Opticlude, Ortopad); the 8 children in the part-time patching group were also prescribed 2 h of daily patching with an adhesive-style patch (i.e., Coverlet, 3M Opticlude, Ortopad). During each visit, VA and CSF were measured in each eye under optimal refractive correction by a certified examiner who was blinded to the treatment allocation by using the ETDRS (Figure 1C). They were evaluated at 1, 3, and 6 months. A study completion flowchart for each treatment group is shown in Figure 1A.

### 2.5. PL Therapy

The patients’ refractive statuses were best corrected 1 month prior to therapy initiation. In each session, an algorithm analyzed the subjects’ responses and accordingly adjusted the visual difficulty to a level that will most effectively result to improvements. The treatment was applied in successive 30-min sessions daily, along with 2 h of patching with an adhesive-style patch (i.e., Coverlet, 3M Opticlude, Ortopad) daily. The sessions in the first week were performed under supervision at the clinic, with additional sessions performed at home. Each treatment station (home PC) was connected to the central database server via the Internet. After each training session, the results were automatically sent to the server via the Internet. In PL, Gabor patches were used as the basic stimuli, which are generated by a cosinusoidal model, and they were enveloped by a stationary Gaussian. The entire stimuli set consisted of one central Gabor target (low-contrast) placed exactly in the fixation area and flanked above and below by two collinear high-contrast Gabor patches. The algorithm fine tunes the distance between the central Gabor and the flankers to maximize the contrast response depending on the participant’s mouse interaction during the training [10,18]. The algorithm, which operates in the central server, calculates the results for a specific patient and sends them back to the appropriate station, and this is a tailored training task.

The PL procedure was as follows: The contrast threshold check procedure was performed by using the 1-up/3-down staircase [19], and then the temporal-2AFC procedure was applied. The stimulus duration varied from 80 ms to 320 ms, starting from the easiest (320 ms); the stimulus duration was reduced depending on the participants’ daily performances. Daily training is terminated after 9 (sessions) × 100 (trials), for a total duration of 30 to 45 min.

The monitor was set up as follows: All stimuli were displayed on an LCD monitor with a refresh rate of 60Hz and a luminance resolution of 8 bits. A calibration was carefully performed prior to each training to ensure the correct presentation of stimuli (size, SF, distance between flankers, and central target) at a viewing distance of 150 cm.

### 2.6. Primary and Secondary Outcomes

The primary outcome was the change in the AULCSF, and the secondary outcomes were the changes in (1) LogMAR VA and (2) CSF cutoff.

### 2.7. Statistical Analysis

For stage I (cross-sectional study), the sample size was calculated based on the primary outcome (i.e., AULCSF). The mean AULCSF was expected to be 0.5 in the control group and 1.0 in the PL group, with a common standard deviation of 0.5. A sample size of 23 per group was required to achieve 90% power at a two-sided significance level of 0.05. For stage II (RCT), the sample size was calculated based on the primary outcome (i.e., AULCSF). The mean AULCSF was expected to be 0.5 in the control group and 1.0 in the PL group, with a common standard deviation of 0.3. A sample size of 8 per group was required to achieve 80% power at a two-sided significance level of 0.05. Thus, a sample size of 24 per group in the first stage was sufficient for the RCT part.

The primary and secondary analyses were based on the intention-to-treat (ITT) principle. The Shapiro–Wilk test and Kolmogorov–Smirnov test were used to evaluate the normality of the continuous variables. Data are presented as means ± standard deviations for normal continuous variables, median (p25, p75) for non-normally continuous variables, and as frequency (percentage) for categorical variables. We compared the proportions of the categorical variables with x^2^ tests. When comparing two groups of data, a Student t test was used if the data were normally distributed; otherwise, the Mann–Whitney U test was used. When comparing three groups of data, ANOVA was used if the data were normally distributed; otherwise, the Kruskal–Wallis test and then the post-hoc Dunnet’s test were used for between-group comparison. Statistical significance was defined as two-sided *p* values of <0.05. All statistical analyses were performed using SPSS version 24.0 (SPSS Inc., Chicago, IL, USA).

## 3. Results

### 3.1. CSF Metrics in the LD and N Groups

The qCSF test showed that the CSF was low among the LD patients, as indicated by the mean AULCSF in Figure 2A. Compared with the N group, the LD group and the optical defocus group showed significantly reduced AULCSF values (1.29 ± 0.27 vs. 0.40 ± 0.05 (*p* < 0.001) and 0.70 ± 0.05 (*p* < 0.001)). The difference in the changes in the AULCSF between the normal and LD groups was 0.89 ± 0.06 (*p* < 0.001), and the difference in the changes in the AULCSF between the LD and optical defocus group was 0.30 ± 0.06 (*p* < 0.001). Moreover, the LD group had a lower cutoff SF (5.38 ± 0.75 cpd) than the optical defocus group (8.81 ± 0.74 cpd, *p* = 0.392) and the normal viewing group (14.81 ± 0.89 cpd, *p* < 0.001) (Figure 2B). The difference between the normal and LD groups in terms of the changes in cutoff SF was 5.93 ± 1.27 (*p* < 0.001).

The cumulative probability distributions of the AULCSF for the LD and N groups are shown in Figure 2C. The log CS was significantly lower in the LD group than in the N group, which was subjected to optical defocusing (*p* < 0.001, *p* < 0.01, *p* < 0.01, *p* < 0.05, *p* < 0.05, and *p* < 0.05 at spatial frequencies of 1, 1.5, 3, 6, 12, and 18 cpd, respectively) (n = 9), as well as in the normal viewing group (both *p* < 0.001) (n = 16) (Figure 2D). For example, at an SF of 1 cpd, the log CS for the LD group was 0.97 ± 0.07; the values for the N group, which was subjected to optical defocusing, and for the normal group were 1.35 ± 0.04 (*p* < 0.001) and 1.55 ± 0.07 (*p* < 0.05), respectively, and the difference between the LD group and the N group was −0.62 ± 0.17 (*p* < 0.001). At 1.5 cpd, the corresponding CS values were 0.89 ± 0.08 vs. 1.23 ± 0.05 (*p* < 0.01) and 1.42 ± 0.08 (*p* < 0.05), and the difference between the LD group and the N group was −0.58 ± 0.17 (*p* < 0.01).

Figure 3A–I illustrate the reduction in AULCSF at each SF (from 0.5 cpd to 18 cpd) for the LD group compared with the N group, which was subjected to optical defocusing, with a simulated reduction in matched BCVA (from logMAR BCVA 1.00 to 0.04). The results for the N group served as a reference AULCSF for the normal population.

### 3.2. Improvement in the CSF in the LD Group

The mean improvements in the CSF were seen in the increase in AULCSF from 0.49 ± 0.15 at baseline to 0.73 ± 0.18 (*p* = 0.32), 0.78 ± 0.21 (*p* = 0.28), 0.80 ± 0.19 (*p* = 0.22), and 1.06 ± 0.20 (*p* < 0.05) at 1 week, 1 month, 3 months, and 6 months, respectively, in the PL group, where the change in the AULCSF from the baseline to 6 months was 0.56 ± 0.25 (*p* < 0.05) (Figure 4B, Table 2). The results showed that the efficacy of the therapy was significantly correlated with treatment duration. Comparatively, the AULCSF improved from 0.50 ± 0.14 at baseline to 0.54 ± 0.13 (*p* = 0.83), 0.55 ± 0.14 (*p* = 0.79), 0.51 ± 0.13 (*p* = 0.97), and 0.49 ± 0.14 (*p* = 0.96) at 1 week, 1 month, 3 months, and 6 months, respectively, in the patching group, where the change in the AULCSF from the baseline to 6 months was 0.01 ± 0.20 (*p* = 0.96), which is not an obvious increase (Figure 4B, Table 2). Moreover, the difference between the PL and patching groups in terms of the changes in the AULCSF from the baseline to 6 months was 0.56 ± 0.20 (*p* < 0.05).

The primary outcome, which is the difference in the changes in the AULCSF between the PL and patching groups from the baseline to 6 months, was 0.56 ± 0.25 (*p* < 0.05). The cumulative probability distributions of the AULCSF for the PL and patching groups from the baseline to 6 months are shown in Figure 4A. The intersection at 50% probability represents an indicative AULCSF value, and the values at baseline and 6 months were 0.35 and 0.81 for the PL group, respectively, and 0.35 and 0.34 for the patching group, respectively. Each patient’s performance in the PL group is shown in Figure 4D–L. None of the fellow eyes showed a decrease in the CSF. During the follow-up at 1, 3, and 6 months after the training completion, no obvious improvement or reduction in the CSF was observed.

### 3.3. Improvement in VA in the LD Group with PL and Part-Time Patching

After the adjustment for the baseline VA, the mean BCVA improved from 0.63 ± 0.11 at baseline to 0.53 ± 0.12 at 1 week, with a difference between the means of 0.10 ± 0.16 (*p* = 0.54) in the PL group, whereas that in the control group improved from 0.67 ± 0.09 to 0.65 ± 0.08 (*p* = 0.90). In the PL group, the mean BCVA increased to 0.49 ± 0.11 (*p* = 0.38), 0.48 ± 0.11 (*p* = 0.33), and 0.32 ± 0.09 (*p* < 0.05) at 1, 3, and 6 months, respectively (Figure 4B, Table 3). At 6 months, the mean BCVA difference was 0.31 ± 0.14 and 0.03 ± 0.12 in the PL and patching groups, respectively. The between-group difference in the changes in VA from the baseline to 6 months was −0.30 (95%CI: −0.46, −0.14, *p* < 0.001). None of the fellow eyes showed a decrease in BCVA, and the refractive degree was nearly stable. During the follow-up at 1, 3, and 6 months after the training completion, no obvious improvement or reduction in the BCVA of the amblyopic eyes was observed.

## 4. Discussion

Little research has been conducted on postoperative visual function evaluation and vision reconstruction. This study has found that LD patients after LKP showed obvious visual impairment in terms of VA and CS compared with the normal children and that PL performed better than traditional patching in improving VA and CSF in amblyopic eyes. To our best knowledge, this is the first PL study to focus on LD children and to demonstrate that PL therapy could improve CSF and VA, making PL a promising treatment option for LD children with amblyopia when conventional treatments fail.

LD patients after keratoplasty exhibited obvious VA and CSF deficits compared with the normal children, and these conditions may have been caused by astigmatism and surgical alterations. Hussein et al. have identified that a preoperative and postoperative astigmatism that is greater than 1.5 D is a risk factor for amblyopia, which may be caused by media opacities, corneal suture, or corneal rejection action [14]. The mean spherical and astigmatic refractive errors in our participants were −3.8 ± 3.7 D and −4.9 ± 2.9 D, respectively, which were deemed to be associated with a high risk of amblyopia; thus, our patients might develop amblyopia. Apart from the VA deficits, an evident CSF impairment was observed in the LD children compared with the normal children. In Nielsen and Hjortdal’s study, the patients underwent posterior LKP and had similar VAs and refractive statuses, with a mean log CS of 1.06 ± 0.25 (vs. 0.49 ± 0.15 in our study) [15,20].

To further evaluate the severity of the visual function deficits in the LD group, we designed an experiment wherein normal children wearing an optical defocus could simulate the VA of the LD groups, and then we compared the contrast sensitivity of the LD and N groups. The results indicated that the LD patients after LKP had a worse visual function than the N group, which had the same visual condition under optical defocus. Optical defocus was used in this study to simulate the visual conditions of the patients with LD, which did not degrade the binocular function and which was the most permissive for binocular function [21]. The LD children exhibited an evident CSF impairment (1–0.04 LogMAR BCVA), especially at high SF values, compared with the corresponding optical defocus group nearly at each SF. Although refractive errors are corrected prior to a visual function evaluation, visual recovery may be dependent on further changes in the subepithelial and host stroma, ultimately leading to light scatter and amblyopia [22]. Management of amblyopia must continue after surgical excision to yield optimal results when or if surgery is performed at a young age [23].

In the therapeutic stage, the CSF, including AULCSF and VA, increased significantly, and the efficacy of the therapy was correlated with treatment duration. However, the CSF did not show an obvious increase in the patching group of LD patients. The term “perceptual learning” describes a process whereby practicing certain visual tasks leads to an improvement in visual performance [21]. Visual performance can be improved through repetitive practice of specific controlled visual tasks. These repetitive tasks initiate neural modifications that can lead to improvements in neuronal efficiency [24,25]. It should be emphasized that the LD patients had not shown any improvement with patching for 6 months; nevertheless, in the PL training, the mean BCVA increased from 0.63 ± 0.11 to 0.32 ± 0.09, with an improvement of 0.31, and the AULCSF in the CS increased from 0.49 ± 0.15 to 1.06 ± 0.20, which is encouraging and reveals a positive trend for the PL group. It is worth noting that the improvement in CS appeared only at the low and middle SF values and not at high SF values. However, middle and high SF values have been documented to be particularly useful for target detection and task identification, even in patients in whom VA is not excellent [25]. We deduced that high SF values require a better VA for recovery [26]. Polat et al. studied adult amblyopic patients who were randomized into either a perceptual vision treatment program or a placebo vision training program for amblyopia [10]. The pretreatment VA in both study groups was 0.42 logMAR VA, an improvement by 2.5 lines in the perceptual vision treatment group, but no improvement was observed in the control group [27]. It is likely that the advantage of PL is its being an active task that subjects must perform, whereas patching is passive.

The present findings regarding the effects of PL on LD children have scientific and clinical implications. First, the logMAR BCVA should be at least 1.00 or better to ensure that the children can clearly recognize the visual target, as persistence of the improved visual function indicated that learning is not just a temporary adaptation but rather a long-lasting change in the visual cortex [28]. Second, attention plays an important role in selecting what we learn and do not learn effectively. Third, strict supervision and follow-up visits were considerably important for ensuring learning efficiency and compliance because the learning process is constant and gradual and interruptions worsen the impression of the visual cortex [29]. Therefore, visual changes were not statistically significant until 6 months. We deduced that there might be a dose–effect relationship between the training times and the changes in VA. No obvious VA improvement was observed until 6 months, which may reflect the long-term consolidation of the positive influence of PL training on VA, as well as on visual function. These phenomena are in good agreement with the findings of Barollo [30].

This study is the first RCT on amblyopia secondary to epibulbar dermoids, wherein the LD patients after LKP showed an evident visual impairment in terms of VA and CS compared with the normal children, even with those having the same visual condition under optical defocus or those having worse VA after a serious ocular disease [31]. Moreover, the CSF and VA increased significantly, and the efficacy of the therapy was correlated with treatment duration. However, the improvement in CS only appeared at the low and middle SF values and not at the high SF values. The major limitations of our study were the small sample size, the relatively short-term follow-up, and the still on-going longer follow-up. Due to pragmatic feasibility, a sham PL task for the control group was not performed. Moreover, we admit that the observed training effects in this study were due to the influences of both the PL training and patching. Therefore, the effects of patching were not entirely ruled out. Further investigations involving more subjects and training alone (without patching) are warranted. Last, only Chinese children were included; thus, the conclusions must be validated in other ethnicities.

In summary, children suffering from LD with amblyopia exhibited CSF deficits and VA loss simultaneously. PL might be a good treatment option to improve VA and CSF in these patients.

## Figures and Tables

**Figure 1 jcm-11-01879-f001:**
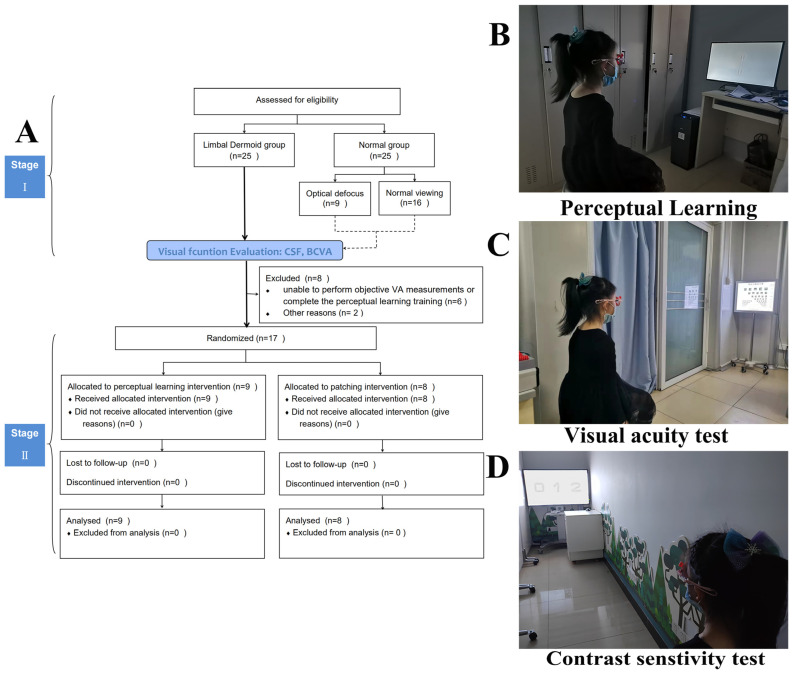
Consolidated Standards of Reporting Trials (CONSORT) Diagram (**A**). The children in the limbal dermoid group underwent perceptual learning daily (**B**), and their contrast sensitivity function and best corrected visual acuity were assessed with quick contrast sensitivity function assessment (**C**) and Early Treatment Diabetic Retinopathy Study (EDTRS) (**D**), respectively.

**Figure 2 jcm-11-01879-f002:**
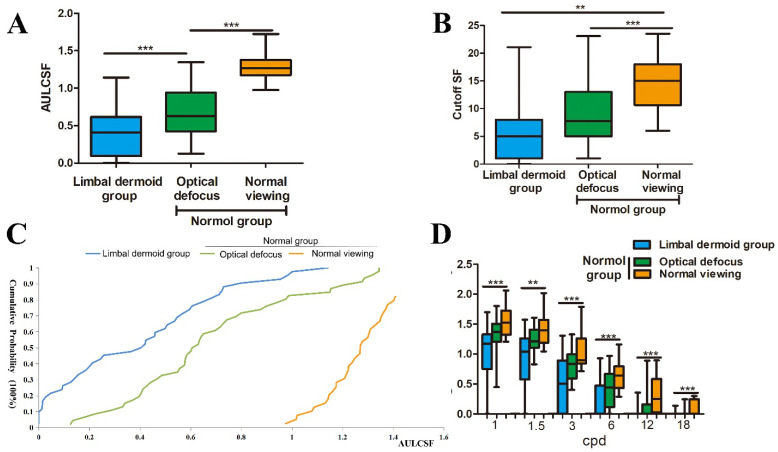
Contrast sensitivity function (CSF) metrics in the limbal demoid (LD) group. Summary of quick CSF statistics for the LD and normal (N) groups. (**A**) A comparison of the area under log CSF (AULCSF), where a value of <1.0 indicates a reduction from the normal viewing range. (**B**) Comparison of the cutoff spatial frequencies (SFs) among the groups. A significant reduction from the cutoff SF for normal viewing conditions was noted. (**C**) A cumulative probability distribution of the AULCSF is projected for the LD and N groups. The intersection at 50% probability represents an indicative AULCSF value, and the values were 0.41 (LD group), 0.68 (plus lens-induced blur viewing group), and 1.27 (normal viewing group). (**D**) Between-group comparison of log contrast sensitivities at 1, 1.5, 3, 6, 12, and 18 cpd. A reduction in CSF could be observed across the SF values tested. **, *p* < 0.01; ***, *p* < 0.001.

**Figure 3 jcm-11-01879-f003:**
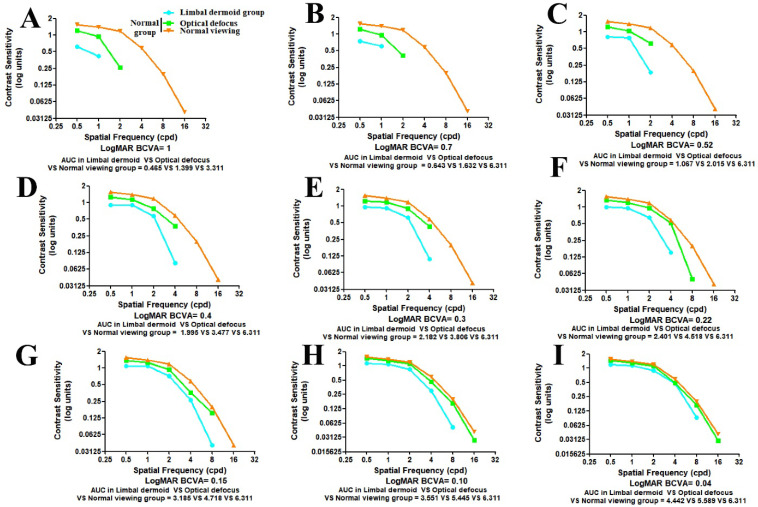
Comparison of the contrast sensitivity function (CSF) metrics for the limbal dermoid (LD) and normal groups. The contrast sensitivity (CS) values increased from BCVA logMAR 1.00 to 0.04 in the LD group and optical defocus group under simulated reduced BCVA conditions. Normal viewing with normal BCVA was plotted as the reference range. A reduction in the area under the CS curve of the LD group relative to that of control subgroups 1 and 2 was noted ((**A**) A_@logMAR_ = 1.00: 0.47 vs. 1.40 and 3.31; (**B**) B_@logMAR_ = 0.70: 0.64 vs. 1.63 and 6.31; (**C**) C_@logMAR_ = 0.52: 1.07 vs. 2.03 and 6.31; (**D**) D_@logMAR_ = 0.40: 2.00 vs. 3.48 and 6.31; and (**E**) E_@logMAR_ = 0.30: 2.18 vs. 3.81 and 6.31); (**F**) F_@logMAR_ = 0.22: 2.40 vs. 4.52 and 6.31; (**G**) G_@logMAR_ = 0.15: 3.19 vs. 4.72 and 6.31; (**H**) H_@logMAR_ = 0.10: 3.55 vs. 5.45 and 6.31; and (**I**) I_@logMAR_ = 0.04: 4.44 vs. 5.59 and 6.31).

**Figure 4 jcm-11-01879-f004:**
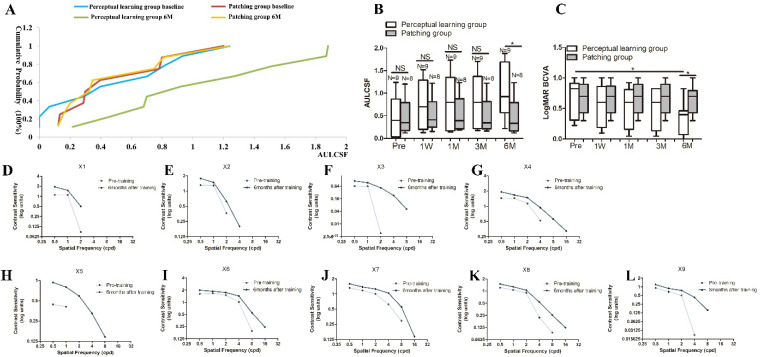
Contrast sensitivity function (CSF) and best corrected visual acuity (BCVA) improvement in the limbal dermoid (LD) patients under perceptual learning (PL). A cumulative probability distribution of the area under log CSF (AULCSF) is projected for the LD and normal groups (**A**). AULCSF and LogMAR BCVA at baseline; 1 week; and 1, 3, and 6 months in the PL and patching groups (**B**,**C**). Each patient’s CSF performance in the PL group is shown in (**D**–**L**). *, *p* < 0.05; ns, not significant.

**Table 1 jcm-11-01879-t001:** Summary of the Demographic Characteristics of the Limbal Dermoid (LD) and Normal (N) Groups.

	LD Group (n = 25)	N Group (n = 25)	*p*
Sex Female, n (%)	12 (48.00)	12 (48.00)	0.99 *
Age (Years)	10.20 (8.30, 11.85)	10.20 ± 1.95	0.87 ^#^
Amblyopic-Eye Spherical Equivalent (Diopters)	−3.86 (−7.75, −0.25)	−4.26 (−5.87, −2.25)	0.899 ^#^
Amblyopic-Eye Cylinder Equivalent (Diopters)	4.86 (4.75, 7.75)	0.30 (0.13, 0.50)	0.000
	Perceptual learning Group (n = 9)	Patching Group (n = 8)	*p*
Sex (Female), n (%)	5 (55.00)	4 (50.00)	0.67 *
Age (Years)	9.31 (7.45, 11.65)	10.40 ± 2.00	0.63 ^#^
Amblyopic-Eye Spherical Equivalent (Diopters)	−2.86 (−8.00, −0.25)	−3.97 (−7.69, −0.31)	0.69 ^#^
Amblyopic-Eye Cylinder Equivalent (Diopters)	4.36 (2.63, 6.63)	4.34 ± 2.96	0.92 ^#^

SD = standard deviation. * Chi-squared test. ^#^ Mann−Whitney U test.

**Table 2 jcm-11-01879-t002:** Comparison of the AULCSF of the Changes in Contrast Sensitivity Function in the PL and Patching Groups (n = 17).

Group	Baseline	Follow-Up Visit at 6 Months	Changes (95%CI) *p*	Between-Group Difference in the Changes (95%CI) *p*
PL group	0.49 ± 0.15	1.06 ± 0.20	0.56 ± 0.25 (*p* < 0.05) *	0.30 ± 0.07 (*p* < 0.001) ^#^
Patching group	0.50 ± 0.14	0.49 ± 0.14	0.01 ± 0.20 (*p* < 0.05) *	

AULCSF = area under log CSF; CI = confidence interval; PL = perceptual learning. * Paired *t*-test. ^#^ Two-sample independent *t*-test.

**Table 3 jcm-11-01879-t003:** Comparison of the LogMAR Visual Acuity in the PL and Patching Groups (n = 17).

Group	Baseline	Follow-Up Visit at 6 Months	Changes (95%CI) *p*	Between-Group Difference in the Changes (95%CI) *p*
PL group	0.63 ± 0.32	0.32 ± 0.64	−0.31 ± 0.13, *p* < 0.05 *	−0.28 ± 0.05, *p* < 0.001 ^#^
Patching group	0.67 ± 0.25	0.64 ± 0.23	−0.02 ± 013, *p* = 0.80	

logMAR = Logarithm of the minimum angle of resolution; CI = confidence interval; PL = perceptual learning. * Paired *t*-test. ^#^ Two-sample independent *t*-test.

## Data Availability

All data will be available upon making a reasonable request to the corresponding author, and they will be shared according to the standards of ethical policies regulating the sharing of data of human subjects.

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
