# Peer review of "Effects of Perceptual Learning on Deprivation Amblyopia in Children with Limbal Dermoid: A Randomized Controlled Trial"

_jcm, 2022, doi:10.3390/jcm11071879_

Round 1

Reviewer 1 Report

  1. Line 81: “…focused on the limbal dermoid children..”

I would write “children with limbal dermoid” instead.

  1. Line 96, 97. “.. the LD patients were randomly assigned to receive with PL combined with or only part time patching therapy”

The sentence is not clear. Please define what types of therapy you mean.

  1. Line 176-178: “The primary outcomes were …”

How many outcomes did you define as primary? 1 outcome only? If yes, please change to outcome (no plural)

Later, in line 285: one outcome.

  1. Line 178: “The secondary outcome were as follow: 1)the change of LogMAR VA from baseline to 6 months”

How many outcomes? If more than one, what were the other results?

  1. Line 224: “Compared with the normal group, the LD group and the normal group who accepted the optical defocusing showed significant reductions in AULCSF values”

The sentence, to be exact, the two compared groups are difficult to see from the sentence. Please define which groups you are comparing and which results were better.

  1. Line 328: “after keratoplasty”

Please always define keratoplasty type.

  1. Line 380-387: The sentence is too long and thus difficultly understandable.

It can be written in 2 sentences instead of 1 long one.

  1. Spacing: In several places (line 32, 37, 39 etc) the spacing between words is missing. This may be not the most important for the manuscript still it makes the text easily readable.

Author Response

Dear Editor,

Thank you very much for giving us an opportunity to revise our manuscript entitled Effects of perceptual learning on deprivation amblyopia in children with limbal dermoid: a randomized controlled trial(jcm-1586560). We also appreciate the editor and reviewers very much for their constructive feedback and insightful suggestions on our manuscript.

We have carefully studied the comments of the reviewers and tried our best to make the revisions accordingly. Attached please find the revised version, which we would like to submit for your consideration. All changes are highlighted in blue( about the reviewer’s suggestion) and red (about the grammar ), so that they may be easily identified. In addition, our point-by-point responses to the comments and suggestions are listed below this letter.

We would like to express our great appreciation to you and reviewers for comments on our paper. Looking forward to hearing from you.

Thank you and best regards.

Yours sincerely,

Jinrong Li

State Key Laboratory of Ophthalmology, Zhongshan Ophthalmic Center, Zhongshan School of Medicine, Sun Yat-Sen University, Guangzhou, China 510064

Phone and Fax: 86-20-8525 3133

Email: lijingr3@mail·sysu·edu·cn

Jin Yuan

State Key Laboratory of Ophthalmology, Zhongshan Ophthalmic Center, Zhongshan School of Medicine, Sun Yat-Sen University, Guangzhou, China 510064

Phone and Fax: 86-20-8525 3133

E-mail: yuanjincornea@126·com

Responses to Reviewer 1:

  1. Line 81: “…focused on the limbal dermoid children..” I would write “children with limbal dermoid” instead.

Response:Thank you very much for your suggestion. Accordingly, this sentence has been revised to “children with limbal dermoid” (Page4 Line78).

  1. Line 96, 97. “.. the LD patients were randomly assigned to receive with PL combined with or only part time patching therapy”. The sentence is not clear. Please define what types of therapy you mean.

Response:Thank you very much for your suggestion. We have detailed the therapy methodology in the revised manuscript. The sentence has been revised to “the LD patients were randomly assigned to receive PL training combined with part-time patching or part-time patching therapy alone” (Page5 Line 116 to 118).

  1. Line 176-178: “The primary outcomes were …”

How many outcomes did you define as primary? 1 outcome only? If yes, please change to outcome (no plural)

Later, in line 285: one outcome.

Response: Thank you very much for pointing out this important issue. There was only one primary outcome, which was the change of the area under log CSF(AULCSF). We have specified this point in the revised manuscript (Page9 Line 216 to 218).

  1. Line 178: “The secondary outcome were as follow: 1)the change of LogMAR VA from baseline to 6 months”

How many outcomes? If more than one, what were the other results?

Response: Thank you for your reminder. There were two secondary outcomes: 1) the change of LogMAR VA; 2) the change of cutoff of CSF. And these contents have been updated, please see Page9 Line 217 to 218 for further details.

  1. Line 224: “Compared with the normal group, the LD group and the normal group who accepted the optical defocusing showed significant reductions in AULCSF values”

The sentence, to be exact, the two compared groups are difficult to see from the sentence. Please define which groups you are comparing and which results were better.

Response: Thank you for your valuable advice. This sentence has been revised to “Compared with the N group, the LD group and the optical defocus group showed significantly reduced AULCSF values...” (Page10, Line 245 to 246).

  1. Line 328: “after keratoplasty”

Please always define keratoplasty type.

Response: Thanks for your comment. All the keratoplasty types have been defined, please see Page4 Line79, Page 5 Line 120, Page 7 Line 162, Page 15 Line 349, Page 16 Line 363, 369, Page 18 Line 413 for further details

  1. Line 380-387: The sentence is too long and thus difficultly understandable.

It can be written in 2 sentences instead of 1 long one.

Response: Thank you very much for giving us valuable suggestion. These sentences have been shortened to make them understandable more easily. Please see Page 17 Line 399 to Line 411 for further details.

  1. Spacing: In several places (line 32, 37, 39 etc) the spacing between words is missing. This may be not the most important for the manuscript still it makes the text easily readable.

Response: Thank you for your reminder. Accordingly, we have invited two colleagues who are skilled authors of English language and an expert statistical scientist to revise the manuscript.  The revised manuscript has been edited and proofread by a medical editing company (http://www.scribendi.com/). We believe that the manuscript has been significantly improved and hope it meets your requirement.

Reviewer 2 Report

A great deal of research has been conducted on perceptual learning and its potential benefits in amblyopia, and this is an important area. However, results have not demonstrated clear benefits and meta-analyses have shown that improvements may be no higher than test-retest differences. However, this study seems to demonstrate clinically significant improvements in acuity and contrast sensitivity over a six month period. If these improvements were due to the perceptual learning the findings are important. However, it is unclear whether the differences might reflect levels of compliance or a placebo effect (since the control group did not undergo similar ‘sham’ training). In addition, the authors’ meaning throughout the manuscript is obscured by problems in the writing including grammar and expression. Significant issues requiring attention are as follows:

In the main text Methods section, the authors rightly state that masking is not possible due to the type of intervention, but it would be possible to have a sham PL task in the control group. It is unclear why this type of control was not used and this should be discussed, explaining why the type of control was selected and whether the lack of placebo control may have affected the results.

The writing is problematic throughout the manuscript. One example is ‘At each visit, VA was measured in each eye with optimal refractive correction by a mask study-certified examiner using the ETDRS(Figure 1C), and the CSF to start of trail, all the children completed training sessions. And they were evaluated at 1st, 3rd and 6th months.’ The meaning of mask study-certified is unclear, as is ‘CSF to start of trail [presumably trial]’ and the second sentence could be part of the first. Another example is ‘which two groups had similar sphere diopters while the LD group showed more cylinder diopters about 4D obviously than the N group’. Numerous sentences are similarly unclear and unfortunately much of the meaning is lost in the writing throughout the manuscript.

The algorithm used to set visual difficulty level in the PL therapy needs to be more fully explained. For example, the ‘range most effective for further improvement’ is not clear. It is stated that the child was patched for 2 hours per day, but it is unclear whether this included the 30-minute PL training period. Presumably so, but this needs to be explicit.

The PL protocol itself is not clear. The task should be explained in detail including the type of stimulus, type of task, viewing distance and temporal parameters. This is important because previous PL studies have been equivocal on whether the learning occurs at only the trained parameters or extends to other parameters.

The simulated reduction in matched BCVA used to construct figure 3 does not seem to be explained in Methods, and should be.

Figure 4B and the text on page 14 show very little change in BCVA from month 1 to month 3. This seems surprising given that an improvement occurs later, at month six. The authors should consider and discuss why the acuity seems to stall for a significant period during training, then improves. In the Discussion it is stated that the follow up visits reinforced the learning process and ‘therefore statistical significance on visual improvement was not observed until 6 months in our study’ but the link between the follow ups and the lack of improvement until 6 months does not seem clear.

The children underwent PL in the clinic initially (at week 1) then at home. Presumably compliance could be checked via the data transmitted after each training session. The children undergoing patching without PL were patched at home. There is no explanation of whether or how compliance with either protocol was monitored. In the Discussion, the authors state that the follow up visits aided compliance, but it is not clear whether there was any monitoring. This is important, since previous research has shown that compliance is about 50% on average (Wallace et al, 2013), so children may patch for about half the prescribed period of time.

Relatively minor points are as follows:

Subject numbers are confusing in the Abstract Methods section (changing from 25 to 17 in the LD group without apparent explanation), and are not clear until figure 1 is shown in the main text. These numbers need to be clarified in the Abstract.

In section 2.5, only one secondary outcome is listed, so the number 1 is not needed.

On page 14 it is stated that the VA in the fellow eye was the same as that in the better eye, referring to figure 4B and table 3. However the figure/table does not show fellow eye or better eye data (readers might interpret the fellow eye as being the same as the better eye, since ‘fellow eye’ is a time widely used to describe the better eye in amblyopia). The authors need to clarify the point being made here.

The text on page 13 states that figure 4B shows cumulative probability distributions of the AULCSF’ but in fact this figure seems to show BCVA as a function of time.

The term ‘difference in changes of…improvement’ is used to describe differences between the contrast sensitivity improvements in two groups. However the use of both ‘changes’ and ‘improvement’ as well as ‘difference’ makes this confusing. As explained above, the wording including grammar and expression should be improved to ensure that the meaning is clear.

Author Response

Dear Editor,

Thank you very much for giving us an opportunity to revise our manuscript entitled Effects of perceptual learning on deprivation amblyopia in children with limbal dermoid: a randomized controlled trial(jcm-1586560). We also appreciate the editor and reviewers very much for their constructive feedback and insightful suggestions on our manuscript.

We have carefully studied the comments of the reviewers and tried our best to make the revisions accordingly. Attached please find the revised version, which we would like to submit for your consideration. All changes are highlighted in blue( about the reviewer’s suggestion) and red (about the grammar ), so that they may be easily identified. In addition, our point-by-point responses to the comments and suggestions are listed below this letter.

We would like to express our great appreciation to you and reviewers for comments on our paper. Looking forward to hearing from you.

Thank you and best regards.

Yours sincerely,

Jinrong Li

State Key Laboratory of Ophthalmology, Zhongshan Ophthalmic Center, Zhongshan School of Medicine, Sun Yat-Sen University, Guangzhou, China 510064

Phone and Fax: 86-20-8525 3133

Email: lijingr3@mail·sysu·edu·cn

Jin Yuan

State Key Laboratory of Ophthalmology, Zhongshan Ophthalmic Center, Zhongshan School of Medicine, Sun Yat-Sen University, Guangzhou, China 510064

Phone and Fax: 86-20-8525 3133

E-mail: yuanjincornea@126·com

Comments and Suggestions for Authors

A great deal of research has been conducted on perceptual learning and its potential benefits in amblyopia, and this is an important area. However, results have not demonstrated clear benefits and meta-analyses have shown that improvements may be no higher than test-retest differences. However, this study seems to demonstrate clinically significant improvements in acuity and contrast sensitivity over a six month period. If these improvements were due to the perceptual learning the findings are important. However, it is unclear whether the differences might reflect levels of compliance or a placebo effect (since the control group did not undergo similar ‘sham’ training). In addition, the authors’ meaning throughout the manuscript is obscured by problems in the writing including grammar and expression. Significant issues requiring attention are as follows:

Response: Authors would like to thank Reviewer #2 for the considering our manuscript interesting and reconsideration after revisions. Thank you very much for your very kind review and giving us constructive suggestions. We have incorporated the following specific comments in preparation of revised version of manuscript.

  1. In the main text Methods section, the authors rightly state that masking is not possible due to the type of intervention, but it would be possible to have a sham PL task in the control group. It is unclear why this type of control was not used and this should be discussed, explaining why the type of control was selected and whether the lack of placebo control may have affected the results.

Response: Thank you for pointing this important issue. We totally agree with you that the masking and truly placebo group were essential for a pragmatic RCT study. For this point, adding a control group with a sham PL task obviously would make the study more rigorous. Before this RCT, we have tried a pilot study to achieve this target by conducting a sham invention in volunteers. However, this procedure led to complaints, unwellness to follow-up, worries for diseases progression. Considering these issues of feasibility in Chinese population, we finally to design a RCT without sham-group but alternatively non-sham PL intervention. Similarly, a recent RCT in our institute, which compared light device + glass versus glass for treatment myopia also failure to set a sham due to feasibility. We accepted your comments. Accordingly, we have stressed this point in the Limitation sections: Due to pragmatic feasibility, we did not implement sham task as the placebo group. We have also toned down the conclusions to make more conservative interpretation. Hopefully these addresses your concerns. (Page 18; Line 419 to 421)

[1] Mingguang He, Effect of Repeated Low-Level Red-Light Therapy for Myopia Control in Children: A Multicenter Randomized Controlled Trial. Ophthalmology. 2021 Dec 1;S0161-6420(21)00916-7.

  1. The writing is problematic throughout the manuscript.

One example is ‘At each visit, VA was measured in each eye with optimal refractive correction by a mask study-certified examiner using the ETDRS(Figure 1C), and the CSF to start of trail, all the children completed training sessions. And they were evaluated at 1st, 3rd and 6th months.’

Response: Thanks for your comment. The manuscript has been rephrased to make more readable. The revised manuscript has been edited and proofread by a medical editing company (http://www.scribendi.com/). We have revised this sentence, and have asked the skilled authors of English language papers to help us for checking the grammar mistakes, please see the revised files for further details.

  1. The meaning of mask study-certified is unclear, as is ‘CSF to start of trail [presumably trial]’ and the second sentence could be part of the first. Another example is ‘which two groups had similar sphere diopters while the LD group showed more cylinder diopters about 4D obviously than the N group’. Numerous sentences are similarly unclear and unfortunately much of the meaning is lost in the writing throughout the manuscript.

Response: Thanks for your comment. We have deleted some unclear sentences, such as “CSF to start of trail” according to your valuable suggestion and revised the sentence you mentioned “..., which the LD group had more cylinder diopters about 4D obviously than the N group.”. Please see Page 7 Line 177 to 180 for further details. Moreover, we have asked the skilled authors of English language papers to help us for checking the grammar mistakes, please see the revised files for further details.

  1. The algorithm used to set visual difficulty level in the PL therapy needs to be more fully explained.

Response: Thanks for your comment.

PL were using Gabor patches as the basic stimuli, generated by a cosinusoidal model, and enveloped by a stationary Gaussian. The whole stimuli set consisted of 1 central Gabor target (low contrast) placed exactly in the fixation area, and flanked above and bellow by two collinear high contrast Gabor patches. The algorithm fine tune the distance between central Gabor and flankers to maximize contrast-response (Polat et al., 2004 ;Polat, 1998;), according to the participant’s mouse interaction during training. And these descriptions have been added in the methods part, please see Page9 Line197 to 205 for further details.

  • Polat U, Ma-Naim T, Belkin M, Sagi D. Improving vision in adult amblyopia by perceptual learning. Proc Natl Acad Sci U S A. 2004 Apr 27;101(17):6692-7.

[2]U Polat, K Mizobe, M W Pettet, T Kasamatsu, A M Norcia. Collinear stimuli regulate visual responses depending on cell's contrast threshold. Nature, 1998 Feb 5;391(6667):580-4. doi: 10.1038/35372.

  1. For example, the ‘range most effective for further improvement’ is not clear. It is stated that the child was patched for 2 hours per day, but it is unclear whether this included the 30-minute PL training period. Presumably so, but this needs to be explicit.

Response: Thank you for your reminder. These sentences have been revised to “In each session, an algorithm analyzed the subject’s responses, and accordingly adjusted the level of visual difficulty to achieve most effective for further improvement.” In the PL group, it was patched for 2.5 hours.

We had to admit that the observed training effects in the current study were due to the influences of both training and patching. Therefore, the effects of patching were not entirely ruled out in this study. Further investigations with more subject and only training (no patching) are necessary.

We have stressed this point in the revised Limitation section, please see Page 18 Line 420 to 423 for further details.

  1. The PL protocol itself is not clear. The task should be explained in detail including the type of stimulus, type of task, viewing distance and temporal parameters. This is important because previous PL studies have been equivocal on whether the learning occurs at only the trained parameters or extends to other parameters.

Response: We thank you for kindly raising this insightful suggestion.

PL were using Gabor patches as the basic stimuli, generated by a cosinusoidal model, and enveloped by a stationary Gaussian. The whole stimuli set consisted of 1 central Gabor target (low contrast) placed exactly in the fixation area, and flanked above and bellow by two collinear high contrast Gabor patches. The algorithm fine tune the distance between central Gabor and flankers to maximize contrast-response (Polat et al., 2004 ;Polat, 1998;), according to the participant’s mouse interaction during training.

PL procedure:

The contrast threshold check procedure were done by using 1-up/3-down staircase(Levit, 1971), and apply the temporal-2AFC procedure. The stimulus duration

varied from 80 to 320 ms, start from easier 320 ms, and reduce the stimulus duration depends on the participant’s daily performance. Daily training terminated after 9 (sessions) * 100 (trails), approximately 30-45 minutes.

Monitor:

All stimuli were displayed on a LCD monitor with refresh rate of 60Hz, and 8 bits of luminance resolution. The calibration was carefully done before each training started, to ensure correct presentation of stimuli (size, spatial frequency, distance between flankers and central target) at the viewing distance of 150cm.

We have detailed the design, intervention, compliance, data monitoring in the supplementary protocol. Hopefully this addresses your concerns.

[1]Polat U, Ma-Naim T, Belkin M, Sagi D. Improving vision in adult amblyopia by perceptual learning. Proc Natl Acad Sci U S A. 2004 Apr 27;101(17):6692-7.

[2]U Polat, K Mizobe, M W Pettet, T Kasamatsu, A M Norcia. Collinear stimuli regulate visual responses depending on cell's contrast threshold. Nature, 1998 Feb 5;391(6667):580-4. doi: 10.1038/35372.

[3]Levitt H. Transformed up-down methods in psychoacoustics. J Acoust Soc Am. 1971 Feb;49(2):Suppl 2:467+.

  1. The simulated reduction in matched BCVA used to construct figure 3 does not seem to be explained in Methods, and should be.

Response: Thank you for pointing out our negligence.

All the children would accepted the VA and CSF measurement under two conditions in normal group: one was under full refractive correction, and another was under full refractive correction plus optical defocus (+1·00 D to +6·00 D positive spherical lens) on the right eye with the vision of the left eye unaltered.[1,2] Specifically, optical defocusing was used to simulate monocular blurred BCVA in the LD group. For example, five patients had a logMAR VA of 0·40; accordingly, the vision of five normal subjects was blurred to a logMAR VA of 0·40 using optical defocus to match the vision of the LD patients while undergoing the qCSF assessment.

[1]Rosser DA, Murdoch IE, Cousens SN. The effect of optical defocus on the test-retest variability of visual acuity measurements. Invest Ophthalmol Vis Sci. 2004 Apr;45(4):1076-9.

[2] David Kordek, Laura K Young , Jan Kremláček. Comparison between optical and digital blur using near visual acuity. Sci Rep. 2021 Feb 9;11(1):3437.  

  1. Figure 4B and the text on page 14 show very little change in BCVA from month 1 to month 3. This seems surprising given that an improvement occurs later, at month six. The authors should consider and discuss why the acuity seems to stall for a significant period during training, then improves. In the Discussion it is stated that the follow up visits reinforced the learning process and ‘therefore statistical significance on visual improvement was not observed until 6 months in our study’ but the link between the follow ups and the lack of improvement until 6 months does not seem clear.

Response: Thanks for your comment. We deduced that it might be dose-effect relationship between the training times and the VA improvement. Not obvious VA improvement was observed until month six, may reflect long-term consolidation of the positive influence of PL training on the visual acuity as well as the visual function. And these phenomenon have a good agreement with Michele Barollo’s study[1]. Besides, it might also suggest that the PL training last for at least 6 months for limbal dermoid children with amblyopia. We have added this content in discussion part, please see Page 17 Line 406 to Line 411 for further details.

[1]Michele Barollo, Giulio Contemori, Luca Battaglini, Andrea Pavan, Clara Casco. Perceptual learning improves contrast sensitivity, visual acuity, and foveal crowding in amblyopia. Restor Neurol Neurosci. 2017;35(5):483-496.

  1. The children underwent PL in the clinic initially (at week 1) then at home. Presumably compliance could be checked via the data transmitted after each training session. The children undergoing patching without PL were patched at home. There is no explanation of whether or how compliance with either protocol was monitored. In the Discussion, the authors state that the follow up visits aided compliance, but it is not clear whether there was any monitoring. This is important, since previous research has shown that compliance is about 50% on average (Wallace et al, 2013), so children may patch for about half the prescribed period of time.

Response: Thank you for your reminder. The compliance of children was monitoring according to predefined protocol. Subjects will need to enter their unique accounts and password details, and log in prior to the commencement of each treatment session. The device is connected to a centralized system via the Internet so that treatment data and compliance can be monitored centrally. Three staff members from the project members will monitor treatment compliance and usage time statistics collected once a week. The staff will remind parents/guardians of children every morning to improve compliance. All the details about improving the compliance have been updated in the protocols; Please see Page9 to Page18 in the protocol for further details.

Relatively minor points are as follows:

  1. Subject numbers are confusing in the Abstract Methods section (changing from 25 to 17 in the LD group without apparent explanation), and are not clear until figure 1 is shown in the main text. These numbers need to be clarified in the Abstract.

Response: Thank you very much for pointing out this important issue. The related contents have been revised. Please see Page 2 Line 36 for further details

  1. In section 2.5, only one secondary outcome is listed, so the number 1 is not needed.

Response: Thank you for your reminder. The secondary outcome have updated, please see Page 9 Line 216 to 218 for further details

  1. On page 14 it is stated that the VA in the fellow eye was the same as that in the better eye, referring to figure 4B and table 3. However the figure/table does not show fellow eye or better eye data (readers might interpret the fellow eye as being the same as the better eye, since ‘fellow eye’ is a time widely used to describe the better eye in amblyopia). The authors need to clarify the point being made here.

Response: Thank you for pointing this mistake. We have deleted this confusing sentence to make it more understandable, please see Page 15 Line 335 to 336 for further details.

  1. The text on page 13 states that figure 4B shows cumulative probability distributions of the AULCSF’ but in fact this figure seems to show BCVA as a function of time.

Response: Thank you for pointing this mistake. We have corrected this confusing sentence to make it more understandable, please see Page 13 Line 308 for further details.

  1. The term ‘difference in changes of…improvement’ is used to describe differences between the contrast sensitivity improvements in two groups.

However the use of both ‘changes’ and ‘improvement’ as well as ‘difference’ makes this confusing.

As explained above, the wording including grammar and expression should be improved to ensure that the meaning is clear.

Response:Thank you for your careful review and very constructive suggestion. We have replaced or deleted some “improvement ” in the manuscript.

The authors apologize for the grammar and syntax errors, which has been corrected in the revised manuscript. The revised manuscript has been edited and proofread by a medical editing company (http://www.scribendi.com/). We also have asked two colleagues who are skilled authors of English language papers to help us for checking the English (see the revised manuscript). We hope that the language is now acceptable for the next review process. We especially thank you for your good comments.

Reviewer 3 Report

The paper needs a significant revision.

  1. Please carefully read journal's guiltiness and correct the paper.
  2. Abstract has to be rewritten. Please include introduction, aim etc.
  3. the title is too long.
  4. section 3.1. should be in material and methods as the first part of this section.
  5. if the authors used non-parametric statistical test (U Mann-Withney), all data has to be presented as a Q1, Q2, Q3, not average and standard deviation. In addition, when you compare more than 2 independed groups (figure 2), you cannot use U Mann-Withney, should Kruskal-Wallis and then Dunnet's test
  6. lines 193- parametric method ???? Did you performed a Shapiro-Wilk test. I would like to see "the raw data" from this test (p-value and histograms) in the Suppl. materials.
  7. results were written not clearly and have to be rewritten.
  8. discussion, please consider to cite https://doi.org/10.1007/s40123-021-00420-8
  9. discussion, please write strengths of the work. In addition, the grup is very small - is it not a limitation? Conclussion is too strong.
  10. in general, discussion is boring and not clearly - has to be improved.
  11. number of references is too low, they ought to be updated.
  12. English correction is necessary.

Author Response

Dear Editor,

Thank you very much for giving us an opportunity to revise our manuscript entitled Effects of perceptual learning on deprivation amblyopia in children with limbal dermoid: a randomized controlled trial(jcm-1586560). We also appreciate the editor and reviewers very much for their constructive feedback and insightful suggestions on our manuscript.

We have carefully studied the comments of the reviewers and tried our best to make the revisions accordingly. Attached please find the revised version, which we would like to submit for your consideration. All changes are highlighted in blue( about the reviewer’s suggestion) and red (about the grammar ), so that they may be easily identified. In addition, our point-by-point responses to the comments and suggestions are listed below this letter.

We would like to express our great appreciation to you and reviewers for comments on our paper. Looking forward to hearing from you.

Thank you and best regards.

Yours sincerely,

Jinrong Li

State Key Laboratory of Ophthalmology, Zhongshan Ophthalmic Center, Zhongshan School of Medicine, Sun Yat-Sen University, Guangzhou, China 510064

Phone and Fax: 86-20-8525 3133

Email: lijingr3@mail·sysu·edu·cn

Jin Yuan

State Key Laboratory of Ophthalmology, Zhongshan Ophthalmic Center, Zhongshan School of Medicine, Sun Yat-Sen University, Guangzhou, China 510064

Phone and Fax: 86-20-8525 3133

E-mail: yuanjincornea@126·com

  1. Abstract has to be rewritten. Please include introduction, aim etc.

Response: Thank you very much for giving us positive comments and valuable suggestions. The abstract has been rewritten, please see Page 2 Line 31 to 43 for further details.

  1. the title is too long.

Response: Thank you for your reminder. We have shorten the title, please see Page 1 Line 1-2 for further details.

  1. section 3.1. should be in material and methods as the first part of this section.

Response: Thanks for your comment. We have moved this part into material and methods. Please see Page 4 Line 84 to Page 5 Line 104 for further details.

  1. if the authors used non-parametric statistical test (U Mann-Withney), all data has to be presented as a Q1, Q2, Q3, not average and standard deviation. In addition, when you compare more than 2 independed groups (figure 2), you cannot use U Mann-Withney, should Kruskal-Wallis and then Dunnet's test. lines 193- parametric method ???? Did you performed a Shapiro-Wilk test. I would like to see "the raw data" from this test (p-value and histograms) in the Suppl. materials.

Response: Thanks for your comment. We apologized for the vague descriptions of statistical analyses. Accordingly, we have re-analyzed the data and updated the methods and results. We compared proportions of categorical variables with x2 tests. When comparing two groups of data, if the data conform to a normal distribution, use the T test to compare between groups, if not, use the Mann-Whitney U test to compare between groups. When comparing the three groups of data, if the data conform to the normal distribution, use ANOVA to compare between groups, if not, use Kruskal-Wallis and then Dunnet's test to compare between groups. The updated results were consistent with our primary analyses (Table 1, 2). Please see Page10 Line230 to 240 for further details.

  1. Results were written not clearly and have to be rewritten.

Response:Thanks for your reminder. We have updated some description of the results part. Please see Page10 Line245 to Page15 Line 344 for further details.

  1. Discussion, please consider to cite https://doi.org/10.1007/s40123-021-00420-8. discussion, please write strengths of the work. In addition, the grup is very small - is it not a limitation? in general, discussion is boring and not clearly - has to be improved.

Response:Thanks for your suggestion. The paper “Krysik K, Miklaszewski P, Dobrowolski D, Lyssek-Boroń A, Grabarek BO, Wylęgała E. Ocular Surface Preparation Before Keratoprosthesis Implantation. Ophthalmol Ther. 2022 Feb;11(1):249-259. ” has been cited. Please see Page18 Line415 for further details. Moreover, the strengths of the work have been emphasized, and the small sample size have been included in the limitation part. Please see Page 18 Line 412 to Line 424 for further details.

  1. number of references is too low, they ought to be updated.

Response:Thank you very much for your suggestion. The references have been updated, please see Page 19 Line 454 to Page 21 Line 514 For further details.

  1. Conclusion is too strong. might be is suggested indicate modify

Response:Thank you very much for your suggestion. We have modified the conclusion to make more conservative statement: In summary, children suffering from limbal dermoid with amblyopia exhibited CSF deficits as well as VA loss simultaneously. PL might be a good choice for limbal dermoid with amblyopia to improve VA and CSF. Hopefully this addresses your concerns. Please see Page 19 Line 425 to 427 for further details.

  1. English correction is necessary.

Response: The authors apologize for the grammar and syntax errors, which has been corrected in the revised manuscript. The revised manuscript has been edited and proofread by a medical editing company (http://www.scribendi.com/). We also have asked two colleagues who are skilled authors of English language papers to help us for checking the English (see the revised manuscript). We hope that the language is now acceptable for the next review process. We especially thank you for your good comments.

Round 2

Reviewer 3 Report

The paper has been improved. The short title ought to be a title of this paper when it is accepted.